# Associations of COVID-19 Knowledge and Risk Perception with the Full Adoption of Preventive Behaviors in Seoul

**DOI:** 10.3390/ijerph182212102

**Published:** 2021-11-18

**Authors:** Jina Choo, Sooyeon Park, Songwhi Noh

**Affiliations:** 1College of Nursing, Korea University, Seoul 02841, Korea; isypark@korea.ac.kr (S.P.); happyronen@korea.ac.kr (S.N.); 2Transdisciplinary Major in Learning Health Systems, Graduate School, Korea University, Seoul 02841, Korea; 3Expert Group on Health Promotion for the Seoul Metropolitan City, Seoul 02841, Korea

**Keywords:** COVID-19, emerging communicable disease, risk reduction behavior, community participation, prevention and control, Seoul

## Abstract

This study explores the levels of COVID-19 knowledge, risk perception, and preventive behavior practice in Seoul, to determine whether knowledge and risk perception are significantly associated with the full adoption of preventive behaviors, for the delivery of a customized public campaign to Seoul’s citizens. A total of 3000 Seoul residents participated in this study through an online questionnaire survey. They had a mean score of 84.6 for COVID-19 knowledge (range: 0–100 points) and 4.2 (range: 1–7 points) for risk perception. Of the participants, 33.4% practiced full adoption of all three preventive behaviors: hand hygiene, wearing a face mask, and social distancing; wearing a face mask was practiced the most (81.0%). Women significantly adopted these three preventive behaviors more often compared with men. Both COVID-19 knowledge and risk perception were found to be significantly associated with the full adoption of preventive behaviors; however, this association differed by the type of preventive behavior. This indicates that city-level information on the levels of COVID-19 knowledge, risk perception, and preventive behaviors should be clearly and periodically communicated among public officers and healthcare professionals to continually raise the public’s awareness of the full adoption of non-pharmaceutical preventive behaviors.

## 1. Introduction

In March 2020, the World Health Organization (WHO) declared the coronavirus disease (COVID-19) a global pandemic and suggested that the world actively respond to its spread [1]. Since the first case was reported in Wuhan, China in December 2019 [2], the cumulative number of confirmed COVID-19 cases had quickly exceeded 241,094,716 in 216 countries as of October 2021 [3]. In the earlier part of the pandemic, the sudden appearance and rapid worldwide spread of COVID-19 caused confusion and anxiety in various countries given the misinformation regarding unclear transmission routes, although the transmission was fairly controlled and its method agreed upon to be human-to-human respiratory droplets or direct contact [4].

Non-pharmaceutical interventions are reportedly the most efficient strategy to block the spread of COVID-19 in the community while waiting to achieve herd immunity from vaccination [5,6]. Several researchers, as well as the WHO, have suggested that non-pharmaceutical preventive behaviors such as hand hygiene, wearing a face mask, and social distancing could slow the spread of infectious diseases [7]. The spread of COVID-19 in East Asian countries, such as Korea and China, was mitigated through the active participation of citizens in non-pharmaceutical interventions, although the infectious disease management policies differ from country to country [8].

Public knowledge and risk perception as well as non-pharmaceutical preventive behaviors may play an important role in controlling the COVID-19 pandemic, as seen in previous cases, such as severe acute respiratory syndrome (SARS), Ebola, and bird flu [9,10,11]. The public’s level of knowledge is an antecedent to adopting preventive behaviors against an infectious disease at the community level because knowledge empowers the public to make appropriate decisions regarding the practice of preventive behaviors [10,12,13]. In the past, the level of knowledge about SARS and H1N1 enhanced the practice of preventive behaviors against these diseases [14,15]. It was recently reported that high levels of COVID-19 knowledge were significantly associated with the practice of preventive behaviors such as hand hygiene and social distancing among specific groups (i.e., college students, medical students, and health care workers) [4,16,17]. Meanwhile, the public’s risk perception is another factor that influences the motivation and willingness to perform non-pharmaceutical preventive behaviors against infectious diseases [18]. During the COVID-19 pandemic, high levels of risk perception were also positively associated with the practice of preventive behaviors such as hand hygiene and wearing a face mask among the Chinese population [19]. Therefore, as the COVID-19 pandemic is prolonged, an accurate assessment of the public’s COVID-19 knowledge and risk perception is necessary to enhance adoption of preventive behaviors and can assist policymakers and healthcare professionals in effectively controlling the spread of COVID-19 [20,21].

The South Korean government’s responses to the COVID-19 outbreak has been considered effective in terms of aggressive epidemiological investigation, rigorous tracing, and rapid mass testing of the potential contacts of confirmed cases [22]. Moreover, non-pharmaceutical preventive measures were also actively introduced in February 2020, specifically, a public and fair distribution of face masks in response to a shortage, and subsequently strict introduction of social distancing after a large-scale spread of COVID-19 resulting from a massive gathering in a local province [22]. The WHO strengthened the protocols for city-level preparedness and heightened local authorities’ governance and policy-making responsibilities against the COVID-19 pandemic. It also addressed four key strategies for a robust response to COVID-19; of the four key strategies, contextually appropriate approaches to public health measures—especially social distancing, hand hygiene, and respiratory etiquette—were listed [23]. In March 2021, the Seoul Metropolitan city government introduced the “Social Distancing Campaign: A Pause!” to remind Seoul’s citizens to practice hand hygiene, wear a face mask, and avoid physical contact [24,25].

People may be exposed to too much information regarding the characteristics and management of COVID-19 through various social media [26,27]. As such, an appropriate level of exposure to accurate information during the pandemic is important to prevent misperceptions about COVID-19 risk and increase compliance with preventive measures. Meanwhile, the three major preventive behaviors, hand hygiene, wearing a face mask, and social distancing should be adopted fully, that is, consistently and without fail, in one’s daily routine. However, most studies have evaluated the presence of preventive behaviors rather than their full adoption [4,16,17,19]. There is little information in population-based studies regarding the full adoption rates of these preventive behaviors during the COVID-19 pandemic [28]. Furthermore, no previous studies have reported the associations of COVID-19 knowledge and risk perception with the full adoption of either each of or all of the three preventive behaviors (i.e., hand hygiene, wearing a face mask, and social distancing).

The present study has two aims. The first is to explore the levels of COVID-19 knowledge, risk perception, and full adoption of hand hygiene, face mask wearing, and social distancing behaviors among Seoul’s citizens. The second is to determine whether COVID-19 knowledge and risk perception are significantly associated with the full adoption of either each of or all three behaviors, for the purpose of effectively delivering information campaigns to enhance COVID-19 preventive behaviors.

## 2. Methods

### 2.1. Study Design and Participants

We undertook a secondary analysis using data of a “parent” study that was performed by the Expert Group on Health Promotion for the Seoul Metropolitan Government [29] and was financially supported by Vital Strategies as a mini-grant project of the Partnership for Healthy Cities. The parent study was conducted through an online survey to identify COVID-19 knowledge, risk perception, and preventive behaviors and to subsequently provide person-centered and customized messages to Seoul’s citizens. The online survey was conducted by developing an Internet interface that contained the questionnaires used in the present study. The parent study took six months, from 7 July to 31 December 2020. The online survey in the parent study was executed for eight days. During the online survey, the average number of confirmed COVID-19 cases (a mean of 66.3 cases in the metropolitan area) did not fluctuate [30].

The participants were 3000 Seoul citizens who were recruited by the online survey company. From the 1,400,000 study subjects available on the online survey company’s database, potential participants were extracted as the first step according to the following eligibility criteria: aged over 19 years, living in Seoul City, not working as healthcare workers, and not having been infected with COVID-19. Subsequently, they were randomly sampled proportionally by gender and age group to be representative of the citizens of Seoul. Finally, 120 citizens (*n* = 120) were recruited from each of the 25 municipal counties in Seoul on a first-come first-served basis.

Participants voluntarily agreed to join the survey and take part in the study; they provided written informed consent after receipt of recruitment information and research participation instructions posted on the online board. The study protocols were approved by the Institutional Review Board of Korea University (No. KUIRB-2020-0215-01), and all procedures followed this board’s ethical standards.

### 2.2. Measures

The present study used data about the participants’ general characteristics, COVID-19 knowledge, risk perception, and preventive behaviors.

General characteristics were sociodemographic characteristics, namely, gender, age, education level (college-educated or not), monthly household income, and employment status. Monthly household income was classified into less than 5,000,000 won and 5,000,000 won or more, according to the median level of the general population in South Korea [31].

COVID-19 knowledge was assessed by a modified instrument, using items extracted from three COVID-19 knowledge questionnaires administered by Azlan et al. (2020), Taghrir et al. (2020), and Hwang et al. (2021) [12,32,33]; it consisted of 17 questions regarding the incubation period of the COVID-19 virus, COVID-19 signs and symptoms, high risk groups, diagnosis, transmission, and non-pharmaceutical prevention. It was a self-reported measure, administered with a 3-point “True/False” format with a “Don’t know” option. The COVID-19 knowledge scores were totaled on a scale of 0 to 17 points and were converted into a scale of 0 to 100 points. The Kuder-Richardson reliability coefficient 20 (KR 20) was 0.69 in the present study.

COVID-19 risk perception was measured using a tool developed by Dryhurst et al. (2020), to investigate participants’ risk perception of the COVID-19 situation experienced in daily life [20]. It consisted of the following six items regarding present levels of worry about the COVID-19 virus, perceived likelihood of his/her (or their family and friends) contracting the virus over the next six months, and perceived seriousness of the COVID-19 pandemic: (1) How worried are you personally about the following issues at present—coronavirus disease/COVID-19? (2) How likely do you think it is that you will be directly and personally affected by the following in the next six months—coronavirus disease/COVID-19? (3) How likely do you think it is that your friends and family in the country you are currently living in will be directly affected by the following in the next six months—coronavirus disease/COVID-19? (4) to (6) How much do you agree or disagree with the following statements?—(4) The coronavirus disease/COVID-19 will affect many people in the country I’m currently living in., (5) I will probably get sick with the coronavirus disease/COVID-19, and (6) Getting sick with the coronavirus disease/COVID-19 can be serious. The first three items were rated on a 7-point scale, and the next three items on a 5-point scale. Scores of the items on the 5-point scale were converted to those of the 7-point scale with a range between 1 and 7 points [20]—the higher the score, the higher the risk perception. Cronbach’s alpha was 0.72 in the previous study [20] and 0.71 in the present study. The COVID-19 knowledge and risk perception scales were originally in English [12,20,32,33]. They were first translated into Korean by researchers and then back-translated into English by a Korean–English bilingual user. Next, we confirmed and consolidated them into their final Korean versions.

Non-pharmaceutical preventive behaviors were measured by five items selected by the authors through a literature review of compliance with existing COVID-19 preventive behaviors. They comprised items indicating the degree of individual adoption of each COVID-19 preventive behavior: two items for hand hygiene (washing hands with water and soap or cleaning hands with a hand sanitizer), one for wearing a face mask, and two for social distancing (avoiding public spaces or places where many people gather or maintaining social distancing (2 m) with others in crowded places). Each question was answered on a 4-point Likert scale, “How often did you practice the following behaviors during the last four months?” with a range of responses (i.e., never, sometimes, often, and always practice). Since strictly following the recommended preventive behaviors of hand hygiene, wearing a face mask, and social distancing is the safest way to contain further spread of the infection [34], we recoded the items as “always” versus “not always” (code 1 = always practice; code 0 = never, sometimes, or often practice).

### 2.3. Data Analysis

All data were analyzed using SPSS 24.0 (SPSS Inc., Chicago, IL, USA). A *p*-value < 0.05 was considered statistically significant. Participants’ sociodemographic characteristics were expressed as numbers (%), and gender differences were analyzed using the chi-square test. Participants’ COVID-19 knowledge and risk perception levels were reported as means (standard deviations [SD]), and gender differences were analyzed using the Mann–Whitney U test and the independent *t*-test, respectively, according to the presence of a normal distribution. The COVID-19 knowledge scores showed a right-skewed distribution, while risk perception scores showed a normal distribution. As such, risk perception levels as well as COVID-19 knowledge scores were categorized into low, middle, and high tertile groups. Knowledge scores of 82.4 and 94.1 fell into the middle and high tertile cut-off scores, while risk perception scores of 39.1 and 4.6 fell into the middle and high tertile cut-off scores.

To identify the association between COVID-19 knowledge and risk perception, logistic regression analysis was conducted for Models 1 and 2. In Model 1, the predictor variables of sociodemographic characteristics (i.e., age, gender, education level, income, and employment status) and the outcome variable of COVID-19 risk perception were included. In Model 2, the predictor variables of the same sociodemographic variables as in Model 1 and COVID-19 knowledge and the outcome variable of COVID-19 risk perception were included. For the outcome variable of COVID-19 knowledge, only Model 1 was conducted. The outcome variables of COVID-19 knowledge and risk perception were re-coded in the logistic regression models into high tertile (code = 1) versus low and middle tertile (code = 0).

To examine the associations among COVID-19 knowledge, risk perception, and the full adoption of preventive behaviors, logistic regression analysis was conducted after adjusting for the variables of sociodemographic characteristics. Each variable of COVID-19 knowledge and risk perception was analyzed as tertiles, with the low-tertile group as the reference group. Missing values were analyzed as they were and naturally excluded according to the circumstances required by the logistic regression models.

All the data in the logistic regression models were expressed as odds ratios (OR) and at 95% confidence intervals (CI).

## 3. Results

### 3.1. Participants’ General Characteristics and Levels of COVID-19 Knowledge and Risk Perception

Participants (*n* = 3000) comprised 1463 men (49%) and 1537 women (51%), with a mean age of 43.6 years, ranging from 19–81 years (Table 1). In terms of education level, 76% of the participants were college graduates, and the proportion of college-educated participants was significantly higher among men compared with women (*p* = 0.008). Of the participants, 46% had a monthly household income of 5 million won [31] or more, and 72% were employed. The proportion of employed participants was significantly higher among men compared with women (*p* < 0.001).

Regarding COVID-19 knowledge, participants obtained a mean score of 84.6 points (range: 0–100 points; Table 1). By tertile, the means were 66.4, 85.8, and 96.3 points for the low, middle, and high tertile groups, respectively. The medians and tertile groups did not differ significantly by gender. The questions for which participants obtained scores below 82.4 points (i.e., the cut-off score of the low-tertile group) were regarding the clinical manifestation and diagnosis of COVID-19 and incorrect prevention and treatment measures (i.e., eating garlic, use of an ultraviolet sterilizer, staying in the sun, or keeping antibiotics at home; see Appendix A).

For COVID-19 risk perception, participants obtained a mean of 4.24 points (range: 1–7 points). Moreover, the risk perception scores did not differ significantly by gender.

### 3.2. Full Adoption of Preventive Behaviors against COVID-19

Approximately 33.4% of Seoul’s citizens practiced full adoption of all three preventive behaviors of hand hygiene, wearing a face mask, and social distancing (Figure 1). Participants practiced full adoption of wearing a face mask the most (81.0%), followed by hand hygiene (66.7%), and social distancing (42.6%). The full adoption of these preventive behaviors significantly differed by gender; women were significantly more likely to practice full adoption of these behaviors (*p* < 0.05).

### 3.3. Factors Associated with COVID-19 Knowledge and Risk Perception

Citizens with a college education (OR = 1.43, 95% CI = 1.143–1.779) and a high monthly income (OR = 1.18, 95% CI = 1.012–1.376) were significantly more likely to have a high tertile level of COVID-19 knowledge than their counterparts (Table 2). Further, based on age, the younger the participant, the higher their COVID-19 risk perception (OR = 0.99, 95% CI = 0.980–0.990 in Model 1; OR = 0.99, 95% CI = 0.980–0.990 in Model 2). College-educated citizens were significantly less likely to have a high tertile level of risk perception than high school-educated citizens (OR = 0.81, 95% CI = 0.658–0.989 in Model 2; OR = 0.80, 95% CI = 0.650–0.989 in Model 2). There was no significant association between COVID-19 knowledge and risk perception in a crude model (data not shown). In Model 2 (Table 2), this non-significance remained apparent even after adjusting for age, gender, education level, employment status, and household income.

### 3.4. Associations of the Full Adoption of COVID-19 Preventive Behaviors with General Characteristics, COVID-19 Knowledge, and Risk Perception

For Seoul citizens, an older age was found to be significantly associated with the full adoption of all three preventive behaviors (OR = 1.01, 95% CI = 1.008–1.020), and this significance was specifically apparent for social distancing (OR = 1.02, 95% CI = 1.009–1.021). Women, compared with men, were significantly more likely to practice full adoption of all three preventive behaviors (OR = 1.54, 95% CI = 1.306–1.806). This significance was apparent across all three preventive behaviors (OR = 2.30, 95% CI = 1.954–2.709 for hand hygiene; OR = 1.94, 95% CI = 1.595–2.362 for wearing a face mask; and OR = 1.22, 95% CI = 1.042–1.417 for social distancing). Employed citizens, compared with unemployed, were less likely to practice full adoption of all three preventive behaviors (OR = 0.79, 95% CI = 0.656–0.939). The significance of this finding was apparent in the full adoption of wearing a face mask (OR = 0.74, 95% CI = 0.584–0.931) and social distancing (OR = 0.78, 95% CI = 0.658–0.929).

There were significant associations between COVID-19 knowledge and the full adoption of preventive behaviors. Citizens with a high level of COVID-19 knowledge were significantly more likely to practice full adoption of all three preventive behaviors than those with a low level of COVID-19 knowledge (OR = 1.35, 95% CI = 1.101–1.652 for all behaviors; OR = 1.48, 95% CI = 1.208–1.808 for hand hygiene; OR = 1.65, 95% CI = 1.305–2.097 for wearing a face mask; and OR = 1.23, 95% CI = 1.011–1.485 for social distancing; (Table 3). Additionally, citizens with a middle level of COVID-19 knowledge were significantly more likely to practice full adoption of hand hygiene (OR = 1.33, 95% CI = 1.087–1.629) and face mask wearing (OR = 1.40, 95% CI = 1.103–1.765) than those with a low level of COVID-19 knowledge. However, there was no significant association between COVID-19 knowledge and the full adoption of social distancing.

Furthermore, there were significant associations between COVID-19 risk perception and the full adoption of preventive behaviors. Citizens with a high level of COVID-19 risk perception were significantly more likely to practice the full adoption of all three preventive behaviors than those with a low level of COVID-19 risk perception (OR = 1.59, 95% CI = 1.310–1.938 for all behaviors; OR = 1.46, 95% CI = 1.194–1.796 for hand hygiene; OR = 1.45, 95% CI = 1.333–1.856 for face mask wearing; and OR = 1.70, 95% CI = 1.405–2.051 for social distancing; Table 3). However, citizens with a middle level of COVID-19 risk perception did not differ significantly in their full adoption of all three behaviors, compared with those with a low level of COVID-19 risk perception. COVID-19 risk perception was not significantly associated with the full adoption of either hand hygiene or face mask wearing, but it was significantly associated with the full adoption of social distancing (OR = 1.50, 95% CI = 1.247–1.810).

## 4. Discussion

This study explored the levels of COVID-19 knowledge, risk perception, and preventive behavior practice in Seoul, to determine whether knowledge and risk perception are significantly associated with the full adoption of preventive behaviors, for the delivery of a customized public campaign to Seoul’s citizens. The citizens of Seoul displayed a high and moderate level of COVID-19 knowledge and risk perception, respectively. Approximately one-third (33.4%) of Seoul’s citizens reported full adoption of all three preventive behaviors: hand hygiene, wearing a face mask, and social distancing. They practiced full adoption of wearing a face mask the most (81.0%), followed by hand hygiene (66.7%), and social distancing (42.6%). COVID-19 knowledge and risk perception were significantly associated with the full adoption of all three behaviors; however, this association significantly differed by the type of COVID-19 preventive behavior.

Seoul’s citizens showed a relatively high level of COVID-19 knowledge, with a median correct response score of 84.6%, compared with the scores reported in 2020, specifically, 91.2% in China [29], 74.5% in Egypt [35], 60.0% in Nepal [36], and 54.9% in Bangladesh [37]. The questionnaires used to measure COVID-19 knowledge differ by country; nevertheless, the high levels of COVID-19 knowledge among the participants in this study may be attributed to two factors. First, more college-educated citizens (i.e., 83.7% college graduates) were recruited for the present study compared with the average of all Seoul citizens (i.e., 56.4% college graduates) [38]. In fact, our data showed significant associations of a college education with COVID-19 knowledge. Second, it is likely that Seoul’s citizens were largely exposed to frequent COVID-19 information released at a metropolitan city level [39]. Assuming that governments’ intensive public health education campaigns via media are effective [40], the present finding may be attributable to the intensive city-wide media campaigns conducted by the Seoul city government to raise public awareness of COVID-19 infection risk, symptoms, and preventive measures [39].

However, some issues need clarification. Participants in the high tertile group, but not the middle tertile group, of COVID-19 knowledge scores, were found to be significantly more likely to practice full adoption of preventive behaviors compared with those in the low tertile group. This result indicates that a high level of COVID-19 knowledge in a given population may be necessary for the full and strict adoption of preventive behaviors to mitigate the spread of COVID-19. In this context, on reviewing the question items that participants were not able to answer correctly and in which they obtained scores lower than the high tertile cut-off score (94.2 points; see bold characters in Appendix A) [29], an advanced understanding of COVID-19 itself (i.e., symptoms, incubation period, and diagnosis) and of incorrect preventive measures (e.g., eating garlic and staying in the sun) is essential and should be promoted robustly. Consistent with previous studies [41,42], here, COVID-19 knowledge was found to be associated with a college education and high-income level. In this context, there should be a focus on socioeconomically vulnerable populations when delivering messages on preventive measures, to increase COVID-19 knowledge levels.

Seoul’s citizens’ risk perception level (i.e., 4.24 points) was found to be lower than those obtained from 10 countries in Dryhurst et al.’s study (i.e., 4.78–5.45 points) [20]. This difference might be attributable to the lower number of confirmed cases at the time this study was conducted (15 months after Dryhurst et al.’s study) when the average daily confirmed COVID-19 cases in Seoul had decreased by 22 cases (5632 cumulated cases) [43]. Based on the data, the Korean government lowered the requirement for social distancing from “Phase 1.5” to “Phase 1” in the metropolitan area [44]. “Phase 1” was given the contained status and stable distancing in daily life, based on the criteria of a weekly average number of less than 100 confirmed cases in the metropolitan area (i.e., Seoul and surrounding cities). “Phase 1.5” was given the status manifesting a start of a local epidemic based on the criteria of a weekly average number of 100 confirmed cases or greater (or 40 confirmed cases or greater in the population group aged 60 or greater) in the metropolitan area [44]. Furthermore, the relatively lower levels of COVID-19 risk perception may be attributable to citizens’ high level of trust in the Seoul Metropolitan government resulting from prior experiences of effective city-level responses to the previous outbreak of the Middle East Respiratory Syndrome (MERS) [45].

We found that approximately one-third of the participants followed full adoption of all the three preventive behaviors, that is, hand hygiene, wearing a face mask, and social distancing. The full adoption rate (33.3%) in the present study was higher than the 25.0% rate in a previous study [34], where Tomczyk et al. (2020) examined the full adoption of preventive behaviors concerning nine official recommendations (i.e., coughing etiquette, avoiding handshakes, touching one’s face, mass events, hand hygiene, ventilation, etc.). We found that Seoul citizens adopted wearing a face mask the most (81.0%), followed by hand hygiene (66.7%), and social distancing (42.6%). Recently, Lv et al. (2021), using a 7-point Likert scale, examined 203 Chinese pharmacy students’ adoption of COVID-19 preventive behaviors recommended by the Chinese government [46]; wearing a face mask was adopted the most (6.9 points), followed by social distancing (6.2 points), and hand hygiene (5.9 points). These findings in South Korea and China may be considered in the light of a prevalent social norm of wearing a face mask among Asian populations [27,47]. The practice of wearing a mask is easily observable by others, and this makes people conscious of other people around them [48]. The Korean government’s compulsory regulation on wearing a mask during the COVID-19 pandemic might have led to establishing a strong social norm, which may explain why this preventive behavior was the one that Seoul’s citizens adopted the most.

Consistent with previous studies [49,50,51], our findings indicate that women are more likely than men to adopt preventive behaviors against COVID-19. Generally, women, compared with men, may have greater motivation to promote health [52], perceive the risk of a disease more deeply, and look into disease information more frequently [18]. Given these results, campaigns to raise public awareness of infectious disease control and preventive behaviors should take such gender differences into account [50].

We found a significant association of COVID-19 knowledge with the full adoption of wearing a face mask among Seoul’s citizens. Similarly, Al-Hanawi et al. (2020) also reported a significant and positive association between COVID-19 knowledge and wearing a face mask among 3388 residents of Saudi Arabia; they emphasized governmental efforts, such as public awareness campaigns, to enhance adoption of preventive behaviors against COVID-19 [53]. The Korea Disease Control and Prevention Agency and local governments have made great efforts to initiate an intensive awareness campaign via various types of mass media. More importantly, the government’s transparency in revealing information and dealing with COVID-19 misinformation might have contributed to the public’s engagement in preventive measures [39].

By contrast, in this study, COVID-19 risk perception was not significantly associated with the full adoption of wearing a face mask. The full adoption of face mask wearing may be explained by the establishment of a social norm and sense of individual responsibility established through governmental regulation, rather than risk perception. Siu (2016) reported that not wearing a face mask led to stigmatization and social isolation during the SARS outbreak and possibly motivated the public to wear a face mask [54]. Regarding the cultural aspect, wearing a face mask indoors and outdoors has been commonly practiced in some Asian countries such as Korea, Japan, China, and Vietnam [55], which may have been influenced by the governments’ strong policy in this regard.

We found a significant association of a high level of COVID-19 risk perception with the full adoption of social distancing among Seoul’s citizens, but such an association was not found for COVID-19 knowledge. Xie et al. (2020) also demonstrated a significant association between COVID-19 risk perception and social distancing behaviors among 317 Chinese residents [56]. However, no previous study has investigated the associations of both COVID-19 knowledge and risk perception with social distancing. Schneider et al. (2021) reported that risk perception was a predictor of health protective behaviors over time during the COVID-19 pandemic, and suggested that several sociocultural and experiential factors are involved in the risk perception of COVID-19, such as people’s prosocial tendencies; experience with the virus; or trust in the government, and science and medical professionals [57]. This report suggests that the practice of social distancing during the COVID-19 pandemic is likely to be influenced by sociocultural factors, but not by objective and cognitive factors, such as the individual’s COVID-19 knowledge. The finding that hand hygiene and wearing a face mask have significant associations with COVID-19 knowledge (but not with risk perception) suggests that COVID-19 knowledge and risk perception may operate differently based on the type of preventive behaviors.

Preventing the spread of COVID-19 at a city level may depend heavily on the citizens’ preventive behaviors against the virus [58]. In this context, the levels of infectious disease knowledge and risk perception, as the associates of preventive behaviors during an infectious disease pandemic, should be assessed at a city-level. Subsequently, such citizens’ levels of infectious disease knowledge and risk perception should be used to develop and deliver customized information to the citizens on raising the levels of preventive actions and behaviors. In this regard, in our parent study [29], the levels of COVID-19 knowledge and risk perception were categorized into four quadrant types (i.e., Type 1: low knowledge and high-risk perception, Type 2: high knowledge and high-risk perception, Type 3: Low knowledge and low-risk perception, and Type 4: high knowledge and low-risk perception) (Appendix A). Furthermore, customized information messages were developed and delivered to the citizens and finally evaluated against their satisfaction levels (Appendix A).

Compared to previous studies, the strength of the present study lies in its coverage of a large population and measurement of the full adoption of all three preventive behaviors. In other words, it first identified the full adoption of three necessary COVID-19 preventive behaviors (i.e., hand hygiene, wearing a face mask, and social distancing) at the population level and explored the associations of COVID-19 knowledge and risk perception with the full adoption of these preventive behaviors. Nonetheless, the present study has some limitations. The cross-sectional data in the present study may not guarantee causal relationships between COVID-19 knowledge/risk perception and preventive behaviors. Moreover, knowledge, risk perception, and preventive behaviors may fluctuate over time during the COVID-19 pandemic. Thus, a longitudinal study design may delineate such variables and help in identifying relevant associations concerning COVID-19 preventive behaviors. Moreover, the questionnaire on COVID-19 knowledge developed by the authors might result in biased findings because it was not evaluated with psychometric properties prior to conducting the present study. However, the Kuder-Richardson reliability coefficient 20 (KR 20) was almost up to the cut-off value (i.e., 0.7). We used the questionnaire on COVID-19 to evaluate the risk perception in three dimensions, namely: worry, likelihood of contracting, and seriousness, as used by Dryhurst et al. [20]. Future studies may include items regarding the dimension of unsafety as validated by Man et al. [59], because the construct of risk perception may cover various dimensions. Finally, our findings may not be generalizable to other ethnic groups because Asians have a high compliance rate with these preventive behaviors [27,47]. However, these findings can serve as reference for designing public campaigns to increase compliance with preventive behaviors against infectious disease pandemics.

## 5. Conclusions

Approximately one-third of Seoul’s citizens reported the full adoption of all three preventive behaviors. COVID-19 knowledge and risk perception were significantly associated with the full adoption of all three behaviors. However, this association significantly differed by the type of COVID-19 preventive behavior.

City-level information on the levels of COVID-19 knowledge, risk perception, and preventive behaviors should be clearly and periodically communicated among public officers and healthcare professionals to continually raise the public’s awareness of the full adoption of non-pharmaceutical preventive behaviors. Moreover, as the associations of COVID-19 knowledge and risk perception with preventive behaviors may differ by the type of preventive behavior, accurate identification of the levels of COVID-19 knowledge and risk perception should precede public campaigns for full adoption of the three core preventive behaviors.

## Figures and Tables

**Figure 1 ijerph-18-12102-f001:**
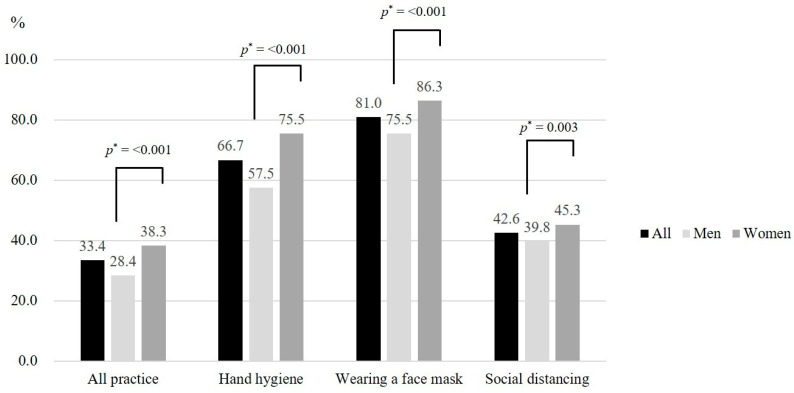
The full adoption rates of non-pharmaceutical preventive behaviors against the COVID-19 pandemic among Seoul’s citizens (*n* = 3000). Notes: Full adoption of the COVID-19 preventive behaviors above was defined as always practicing in daily life. * denotes significance by gender obtained from the chi-square tests.

**Table 1 ijerph-18-12102-t001:** Participants’ general characteristics and levels of COVID-19 knowledge and risk perception (*n* = 3000).

Variables	*n* (%) or Mean (SD)				*p*
	*n*	All	Men (*n* = 1463)	Women (*n* = 1537)	
Age, years	3000	43.6 (14.0)	43.5 (14.1)	43.7 (13.92)	0.779 ^a^
19–29		640 (21.3)	310 (21.2)	330 (21.5)	0.802 ^a^
30–49		1229 (41.0)	611 (41.8)	618 (40.2)	
50–64		914 (30.5)	441 (30.1)	473 (30.8)	
≥65		217 (7.2)	101 (6.9)	116 (7.5)	
Education	3000				0.008 ^a^
>College		2512 (83.7)	1138 (77.8)	1132 (73.6)	
≤High school		488 (16.3)	325 (22.2)	405 (26.4)	
Household income, won	2939				0.083 ^a^
>5,000,000		1360 (46.3)	858 (58.6)	949 (61.7)	
≤5,000,000		1579 (53.7)	605 (41.4)	588 (38.3)	
Employed	2931				<0.001 ^a^
Yes		2118 (72.3)	1151 (78.7)	967 (62.9)	
No		813 (27.7)	312 (21.3)	570 (37.1)	
Knowledge (0–100) ^d^	3000	84.6 (13.52)	84.0 (14.517)	85.1 (12.48)	0.444 ^b^
Low		66.4 (11.77)	64.8 (13.12)	68.0 (10.04)	
Middle		85.8 (2.90)	85.8 (2.90)	85.8 (2.90)	
High		96.3 (2.84)	96.3 (2.84)	96.3 (2.85)	
Risk perception (1–7) ^d^	3000	4.24 (0.82)	4.21 (0.83)	4.26 (0.81)	0.079 ^c^
Low		3.90 (3.36)	3.90 (3.33)	3.90 (3.40)	
Middle		4.60 (4.24)	4.60 (4.24)	4.60 (4.25)	
High		7.00 (5.14)	6.67 (5.14)	7.00 (5.15)	

Notes: SD means standard deviation.^a, b,^ and ^c^ denote significance obtained using the Chi-square, Mann-Whitney U test, and the independent *t*-test; ^d^ denotes the mean (SD).

**Table 2 ijerph-18-12102-t002:** Factors associated with COVID-19 knowledge and risk perception among Seoul’s citizens (*n* = 3000).

Variables	OR (95% CI) ^a^		
	High COVID-19 Knowledge	High COVID-19 Risk Perception	
		Model 1	Model 2
Age	1.01 (1.000–1.011)	**0.99 (0.980–0.990)**	**0.99 (0.980–0.990)**
Women	1.02 (0.873–1.188)	1.08 (0.926–1.251)	1.08 (0.925–1.249)
College-educated	**1.43 (1.143–1.779)**	**0.81 (0.658–0.989)**	**0.80 (0.650–0.989)**
Employed	1.10 (0.918–1.305)	1.18 (0.992–1.395)	1.18 (0.992–1.394)
Household income > 5,000,000 won	**1.18 (1.012–1.376)**	1.01 (0.869–1.173)	1.00 (0.862–1.166)
COVID-19 knowledge			
Low	-	-	ref.
Middle	-	-	1.10 (0.912–1.335)
High	-	-	1.13 (0.937–1.365)

Notes: CI means confidence interval; OR, odds ratio; ref., reference group; and SD, standard deviation. ^a^ is from the logistic regression analysis. Model 1 is the adjusted model for age group, gender, education, income, and employment status; and Model 2, for the variables in Model 1 plus COVID-19 knowledge tertiles. Bold characters indicate significant results.

**Table 3 ijerph-18-12102-t003:** Associations of the full adoption of preventive behaviors with general characteristics, COVID-19 knowledge, and risk perception among Seoul’s citizens (*n* = 3000).

Variables	OR (95% CI)			
	Full Adoption of Preventive Behaviors			
	All Three Behaviors	Hand Hygiene	Face Mask Wearing	Social Distancing
General characteristics				
Age	**1.01 (1.007–1.019)**	1.00 (0.991–1.003)	0.99 (0.986–1.000)	**1.01 (1.009–1.020)**
Women	**1.52 (1.295–1.789)**	**2.29 (1.942–2.690)**	**1.94 (1.591–2.354)**	**1.20 (1.031–1.401)**
College-educated	0.95 (0.767–1.189)	1.03 (0.823–1.292)	0.77 (0.582–1.026)	0.86 (0.693–1.056)
Employed	**0.79 (0.659–0.942)**	0.89 (0.737–1.071)	**0.74 (0.590–0.941)**	**0.78 (0.658–0.929)**
Household income > 5,000,000 won	0.96 (0.819–1.130)	1.01 (0.854–1.183)	0.96 (0.788–1.161)	0.91 (0.784–1.067)
COVID-19 knowledge				
Low	ref.	ref.	ref.	ref.
Middle	1.07 (0.868–1.314)	**1.32 (1.082–1.619)**	**1.39 (1.101–1.759)**	0.96 (0.789–1.166)
High	**1.32 (1.082–1.621)**	**1.47 (1.201–1.793)**	**1.65 (1.302–2.089)**	1.21 (0.998–1.466)
COVID-19 risk perception				
Low	ref.	ref.	ref.	ref.
Middle	0.94 (0.776–1.145)	1.01 (0.837–1.228)	1.00 (0.798–1.258)	0.97 (0.804–1.162)
High	**1.35 (1.111–1.638)**	1.17 (0.956–1.421)	1.09 (0.858–1.375)	**1.50 (1.247–1.810)**

Notes: CI means confidence interval; OR, odds ration; ref., reference group; and SD, standard deviation. Bold characters indicate significant results.

## Data Availability

Not applicable.

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
