# Peer review of "Associations of COVID-19 Knowledge and Risk Perception with the Full Adoption of Preventive Behaviors in Seoul"

_ijerph, 2021, doi:10.3390/ijerph182212102_

Round 1
Reviewer 1 Report
This manuscript aimed to examine to determine whether knowledge and risk perception of COVID-19 are associated with the full adoption of preventive behaviors.
The topic is significant and this manuscript is interesting for readers, but I would like to suggest several corrections.
Major
- Authors asks knowledge of COVID-19 from various points. Firstly, authors extracted questions from previous reports. Is questionnaire consisted of 17 questions validated? If not, authors should mention in the limitation section. Next, does authors think that each knowledge equally contribute to preventive actions? I doubt if diagnosis or symptom did equally affect preventive measure similar to transmission or non-pharmaceutical prevention. All things considered, authors are encouraged to justify the use of this questionnaire.
- I agree that this topic is interesting, but similar studies have been conducted in other countries than Korea. Is the new thing just that this study was conducted in Korea, or if authors can appeal something, it is better to describe something new and important in this study.
- It is better to add some implication on how authors can utilize findings of this study in Seoul citizens or other ones. Currently, I could find description regarding supplementary figures, but I think authors elaborate more about their ideas.
Minor
- Line 104 Authors mentioned that they used data of a "parent" study for the current study. If authors have published any paper describing protocol or main findings of the "parent" study, they should cite it in this manuscript.
- Line 111 This study was conducted for 6 month. It seems that the situation on a degree of spread of COVID-19 was continuously changing. Is it possible that change of the situation in which each responder answered affected the results?
- Line 114 Please clarify the procedure of recruitment of participants. DId authors recruit participants until the number of responders reached 120 in every municipal county? Or did every persons randomly chosen agreed this study? I feel it impossible. If it is not the case, please describe response rate or show the flow of participants.
- Table 1 Knowledge score in Men 14.517 ⇒ 14.52
- Table 1 number of participants are different among items. How did authors deal with missing data. Please clarrify.
- Line 261 The title of table 2 is quite confusing. This table seems to show associated factors of COVID-19 knowledge and risk perception.
- Similarly, In table 2, N=3000. Please explain how did authors deal with missing values.
- Line 311 Since authors did not show data of comparison of outcomes with those in other studies out of Seoul, they can not stress this description.
- Line 320 Since correct response rate of China was higher than Seoul, Korea (authors' study), data of China can be removed from this sentence.
- Line 360, Please add brief explanation how severe "Phase 1.5" and "Phase 1" are.
- Conclusion should include main results, but currently large part of this section is about future perspective.
Author Response
Please see the attached file.
We are very grateful for your comments.
We have revised our manuscript meticulously according to your comments and believe that these changes have made our paper more robust.
The revised statements in the main text are highlighted in red.
Thank you very much!

Reviewer 2 Report
The manuscript describes association between COVID-19 knowledge, risk perception and adoption of preventive behaviors in a large sample of Seoul residents. The paper provides some useful information which may be used for establishing public health campaigns during the COVID-19 pandemic. However, there are some major drawbacks which prevent the manuscript from being published which I have listed below.
Introduction
It generally covers the topic very well, however, I would suggest to reformulate the third paragraph. In lines 57-58 Authors describe the level of knowledge about SARS and H1N1 and applying preventive behaviors. It would be of great value to add information if there were any studies assessing the level of knowledge about COVID-19 and adoption of such preventive behaviors in order to stick to the topic.
Line 92-94: "Furthermore, few studies have reported the associations of COVID-19 knowledge and risk perception with full adoption of the preventive behaviors." - Authors should cite these studies and briefly describe the most important results of the above-mentioned studies, perhaps in the third paragraph.
Methods
Authors must clarify information regarding an online survey - was it performed using Google Forms or something else?
As I understood, Authors based on three different questionnaires assessing the level of COVID-19 knowledge merged them into one questionnaire. I have some doubts regarding the reliability of such tool, as I suppose, it wasn't validated in the studied population.
Moreover, what are the original languages of used questionnaires? How were these questionnaires translated to Korean?
Line 141: The same question regarding the questionnaire developed by Dryhurst et al. - what is the original language of the questionnaire and how was it translated to Korean?
Results
The results are clearly described, however, in Table 1 it is unclear why the column with income is presented as "Income, 10,000 won; > 500; < 500", not simply "Income, > 5,000,000 and < 5,000,000 won
Figure 1: Please correct the font size and font below the figure 1.
Discussion
Within Limitations sub-section Authors should consider issue connected with applied questionnaires which I have given previously.
Author Response

(The authors gave the same response as above.)

Reviewer 3 Report
This work explores the levels of COVID-19 knowledge, risk perception, and
preventive behavior practice in Seoul to determine whether knowledge. This work is interesting and meaningful. Some comments for the authors to improve its quality.
- The research background can be improved by adding more about how risk perception influences people's different behaviors.
- The relationship among the constructs can be investigated using structural equation modeling.
- The randomization of the sampling should be explained clearly.
- There should be a section to explain the practical implications of this work.
- There is a limitation that the dimension of risk perception, i.e. "how safe you personally about the following issues at present" was not investigated. This dimension was emphasized in the work of Man et al., (2019) about the CoWoRP scale. The authors may discuss this and propose a future research opportunity.
Author Response

(The authors gave the same response as above.)

Round 2
Reviewer 2 Report
The Authors of the manuscript have answered to all my concerns sufficiently. Therefore, I find this paper suitable for publication in the IJERPH.
Reviewer 3 Report
The authors did a good job in addressing my comments.